

# The association between serum lipids at diagnosis and renal outcome in microscopic polyangiitis patients

Zigui Zheng[1,2,3,*], Yujia Wang[1,2,3,*], Jingzhi Xie[1,2,3], Zhimin Chen[1,2,3], Bingjing Jiang[1,2,3] and Yanfang Xu[1,2,3]

[1] Department of Nephrology, Blood Purification Research Center, the First Affiliated Hospital of Fujian Medical University, Fuzhou, Fujian, China
[2] Fujian Clinical Research Center for Metabolic Chronic Kidney Disease, the First Affiliated Hospital of Fujian Medical University, Fuzhou, Fujian, China
[3] Department of Nephrology, National Regional Medical Center, Binhai Campus of the First Affiliated Hospital, Fujian Medical University, Fuzhou, Fujian, China
* These authors contributed equally to this work.

Corresponding authors
Yujia Wang,
wangyj2020@fjmu.edu.cn
Yanfang Xu,
xuyanfang99@hotmail.com

## ABSTRACT

**Objectives:** Microscopic polyangiitis (MPA) is a subgroup of ANCA-associated vasculitis (AAV), which is characterized by vascular endothelial cell damage caused by abnormally activated neutrophils. Dyslipidemia is associated with vascular endothelial cell injury, and the relationship between blood lipid levels and renal prognosis in MPA patients is not clear. We aim to investigate the correlation between blood lipid levels at diagnosis and renal prognosis in MPA patients.

**Methods:** Firstly, we retrospectively included 110 patients diagnosed with MPA and the primary endpoint was the occurrence of end stage renal disease (ESRD). The association between blood lipids at diagnosis and renal outcome was evaluated with Cox regression analysis and survival analysis. Secondly, we explored the potential underlying mechanism of poor renal prognosis in patients with high triglycerides (TG) levels at diagnosis using data independent acquisition (DIA) quantitative proteomics.

**Results:** During a median follow-up period of 23 months, 44 out of 110 patients (40%) developed ESRD. High serum TG at diagnosis was associated with ESRD development after adjusting for several confounding factors including age, gender, body mass index (BMI), hypertension, diabetes mellitus, estimated glomerular filtration rate (eGFR) and Birmingham Vasculitis Activity Score (BVAS). Serum very low-density lipoprotein (VLDL) demonstrated a marginal trend towards association with ESRD development. MPA patients with TG >1.45 mmol/L or VLDL > 0.66 mmol/L had significantly higher risk of ESRD development than those with TG ≤ 1.45 mmol/L or VLDL ≤ 0.66 mmol/L. DIA quantitative proteomics analysis suggested that patients with elevated TG levels and severe MPA had an upregulation of profibrotic pathways, inflammatory signaling, and complement and coagulation cascades, in contrast to those with lower TG levels and milder disease severity.

**Conclusions:** In MPA patients, high TG or VLDL at diagnosis is associated with an increased risk of ESRD development. The potential mechanisms may be associated with the upregulation of profibrotic and inflammatory signaling pathways, and the activation of complement and coagulation cascades.

## INTRODUCTION

Antineutrophil cytoplasmic antibodies (ANCA)-associated vasculitis (AAV) is characterized by destruction of small-size vasculitis, with few or no immune complex deposit, which is associated with the presence of serum ANCA targeting neutrophil primary granule proteins (*Jennette, 2013*). AAV can involve small blood vessels in many organs and tissues, giving rise to severe organ-threatening or life-threatening conditions (*Kitching et al., 2020*). The kidney is among the most frequently affected organs in AAV, with renal involvement often manifesting as a swift deterioration in renal function. The prevalence of kidney disease is particularly high, affecting up to 100% of patients with microscopic polyangiitis (MPA) and 70% of those with granulomatosis with polyangiitis (GPA) (*Moiseev et al., 2016*). Approximately 20% of AAV patients will develop end-stage renal disease (ESRD) requiring dialysis or transplantation after immunosuppressive treatment. The mortality rate after ESRD development was reported to be 10.90 deaths per 100 person-years (*Moiseev et al., 2017*). The epidemiological studies of AAV show that the global incidence of AAV has been increasing over the last four decades, with the global incidence of 17.2 per million person-years (*Redondo-Rodriguez et al., 2022*).

The exact cause of AAV remains unknown, but it is widely accepted that the onset of AAV is generally a consequence of the complex interplay between infectious, genetic, and environmental factors (*Ge et al., 2022*; *Kitching et al., 2020*). The activation of neutrophils and the injury of vascular endothelial cells play important roles in AAV pathogenesis (*Ge et al., 2022*). Under the stimulation of pro-inflammatory cytokines such as interleukin-1 (IL-1) and tumor necrosis factor-α (TNF-α), neutrophils display target antigens (such as MPO (myeloperoxidase) or PR3 (Proteinase 3)) on their surface membranes and adhere to the vascular endothelial cells (*Ge et al., 2022*). Exposed autoantigens engage with ANCA, leading to an excessive activation of neutrophils that adhere to endothelial cells. This interaction, in turn, triggers the abnormal production of cytokines, the release of reactive oxygen species (ROS), proteases, and the formation of neutrophil extracellular traps (NETs), ultimately inflicting damage upon the endothelium (*Ge et al., 2022*; *Kronbichler et al., 2020*).

The vascular endothelium is the primary barrier that protects tissues from circulatory invasion (*Liu et al., 2023*). Lipid disturbances are associated with endothelial dysfunction. Hyperlipidemia could mediate endothelial dysfunction by various mechanisms, including the impariment of insulin signaling and nitric oxide production, as well as the exacerbation oxidative stress and inflammation (*Ghosh et al., 2017*). Molecules and signaling pathways such as oxidized low-density lipoprotein (oxLDL), CD36, NF-κB and AMPK/PI3K/Akt/eNOS signaling pathway have been reported to be involved in lipid-mediated endothelial dysfunction (*Goldberg Ira et al., 2021*; *Mahmoud et al., 2017*). Lipid disturbances are also linked with the endothelial stiffness. After binding to CD36, a scavenger receptor, oxLDL can causes endothelial sclerosis by destroying the lipid packaging of the endothelial

membrane and activating the contracted Rho A/ROCK cascade (*Le Master & Levitan, 2019*).

However, there is little evidence on the role of blood lipid levels in renal outcomes in AAV patients. In this study, we aim to investigate the relationship between blood lipids at diagnosis and renal prognosis of MPA, the dominant form of AAV in China, and explore the potential underlying mechanisms by plasma proteomics in a single center study.

## MATERIALS AND METHODS

### Study population

Retrospective part: A total of 110 patients with MPA in the First Affiliated Hospital of Fujian Medical University were retrospectively included from January 2014 to May 2023. The study was in accordance with the provisions of the Declaration of Helsinki and approved by the ethics committee of the First Affiliated Hospital of Fujian Medical University (protocal code [2023]-180, approval 02/28/2023). The patients provided their written informed consent to participate in this study. All patients fulfilled the 2012 Chapel Hill Consensus Conferences Nomenclature of vasculitis and were then reclassified by the 2022 American College of Rheumatology/European Alliance of Associations for Rheumatology Classification Criteria for MPA (*Suppiah et al., 2022*). Exclusion criteria were as follows: (1) died within 3 months of follow-up; (2) had medical history of kidney transplantation; (3) had other comorbid primary kidney diseases; (4) had other comorbid autoimmune diseases; (5) incomplete clinical information in the medical records. The follow-up duration was defined as the period between the date of the diagnosis of MPA and the date of the last visit for survived patients. For deceased patients, the follow-up duration was defined as the period between the initial diagnosis of MPA and the time of death.

Prospective part: Newly diagnosed MPA patients in the First Affiliated Hospital of Fujian Medical University were prospectively enrolled from March 2023 to June 2023. The study was in accordance with the provisions of the Declaration of Helsinki and approved by the ethics committee of the First Affiliated Hospital of Fujian Medical University (protocal code [2023]-180, approval 02/28/2023). The patients provided their written informed consent to participate in this study. Exclusion criteria were as follows: (1) severe infections; (2) had medical history of kidney transplantation; (3) had other comorbid primary kidney diseases; (4) had other comorbid autoimmune diseases; (5) had severe diseases of other important organs including brain, heart, liver and lung; (6) refused to participate.

### Clinical data

The demographic characteristics and clinical data (including medical history of hypertension, diabetes mellitus and chronic kidney disease, laboratory data at diagnosis and during follow-up, histological findings and therapeutic information) of the patients were retrospectively collected from the electronic medical record systems in our hospital. The patients were all screened for laboratory examination at diagnosis including blood lipid levels on admission. The lipids were measured from blood samples collected after an

overnight fast of >8 h, with commercially available test kits and using an automatic chemical analyzer in our hospital. The estimated glomerular filtration rate (eGFR) was calculated by the 2009 CKD (chronic kidney diseases) using the EPI equation. Birmingham vasculitis activity score (BVAS) was used to access the disease activity. The titer of anti-MPO antibody was determined by Enzyme-Linked Immunosorbent Assay (ELISA). Indirect immunofluorescence was used to detect the presence of ANCA. Out of the 110 patients, 48 patients (43.63 %) received a kidney biopsy. The renal biopsy results were determined and scored by two pathologists according to previous published literature (*Aendekerk et al., 2020*). The renal pathology was divided into four types by Berden classification: (1) focal type (≥50% normal glomeruli); (2) Crescentic type (≥50% of crescentic glomeruli); (3) sclerotic type (≥50% globally sclerotic glomeruli); (4) Mixed type (normal, crescent, and sclerosis all exist, but all are less than 50%) (*Bjørneklett, Sriskandarajah & Bostad, 2016*). The therapeutic strategies for MPA at our center were primarily based on the Kidney Disease: Improving Global Outcomes (KDIGO) guidelines. In general, the principles of induction therapy were centered around the combination of cyclophosphamide with glucocorticoids or rituximab with glucocorticoids. For patients with severe organ involvement, the preference leaned towards the combination of cyclophosphamide and glucocorticoids, accompanied by plasma exchange. In a minority of cases where there is no significant organ involvement, mycophenolate mofetil is selected.

## Data independent acquisition quantification proteomics

Blood samples were collected following an overnight fast exceeding 8 h, and plasma was subsequently prepared. For protein extraction, cellular debris in the plasma was initially removed by centrifugation at 12,000 g at 4 °C for 10 min. The supernatant was then transferred to fresh centrifuge tubes. The top 14 high-abundance proteins were depleted using the Pierce™ Top 14 Abundant Protein Depletion Spin Columns Kit from Thermo Fisher Scientific (Waltham, MA, USA). The protein concentration was measured using a BCA kit, following the manufacturer's guidelines. Protein solutions were reduced with 5 mM dithiothreitol for 30 min at 56 °C and alkylated with 11 mM iodoacetamide for 15 min at room temperature in the dark. Alkylated samples were processed through ultrafiltration tubes for FASP digestion. The samples were washed with 8 M urea three times at 12,000 g and room temperature for 20 min each, followed by three washes with 200 mM TEAB. Digestion was carried out overnight with trypsin added at a 1:50 trypsin-to-protein mass ratio. Peptides were recovered by centrifugation at 12,000 g for 10 min, a step repeated twice. The collected peptides were desalted using a Strata X SPE column.

The full MS scan was set at a resolution of 30,000, with a scan range of 390–810 m/z. Precursors were selected for MS/MS analysis using HCD fragmentation at a normalized collision energy (NCE) of 25%, 30%, and 35%. Fragments were detected in the Orbitrap at a resolution of 30,000, with a fixed first mass at 100 m/z. The Automatic Gain Control (AGC) target was set to 3E6, and the maximum injection time was set to auto. DIA data was acquired using the Pulsar Boso alarm in Spectronaut (version 17), with default parameters applied to search the Homo_sapiens_9606_SP_20230103.fasta database,

containing 20,389 sequences. A reverse library was included to calculate the false discovery rate (FDR) due to random matches, with the number of allowed missing sites set to 2 and FDR set at 1%.

For bioinformatics analysis, proteins associated with pathways were extracted from the Kyoto Encyclopedia of Genes and Genomes (KEGG) database. These were intersected with differentially expressed proteomics data to create a protein matrix. Fisher's exact test was applied for GO and KEGG enrichment analysis of the differential proteins. The *P*-value matrix was transformed using a logarithmic scale with base−Log10. Hierarchical clustering analysis was performed on the transformed data using the Euclidean distance metric and average linkage clustering method. The R package "ComplexHeatmap" was utilized to visualize clustering relationships, while the "visNetwork" package was employed to construct a network diagram for the top enrichment results entries.

## Statistical analysis

SPSS 26.0 (IBM, Armonk, NY, USA) and Prism 8.0.2 software were used for statistical analysis. The patients were divided into two groups according to the occurrence of end-stage renal disease or the median of triglycerides (TG)/very low-density liproproteins (VLDL) at diagnosis. Continuous variables are expressed as mean ± SD or as the median with quartile range, and categorical variables were expressed as percentages. Student's t test or Mann-Whitney U test was used for comparison of continuous variables between groups, and Pearson's $\chi^2$ test or Fisher's exact test was used for comparison of categorical variables between groups. Cox regression analysis was used to create a multivariable prediction model for the renal outcome. Receiver operating characteristic (ROC) analysis was used to determine the optimal cut-off value of blood lipids (including TG, total cholesterol (TC), VLDL and low density lipoprotein (LDL)). Kaplan-Meier curve was used to estimate the difference in survival outcomes between groups. Spearman's rank correlation was used to identify the correlation between blood lipid levels at diagnosis and the degree of renal involvement and disease activity. All statistical tests were two-sided tests, and $P < 0.05$ was considered statistically significant.

## RESULTS

### Patient characteristics

A total of 110 MPA patients with 56.4% of male were recruited to this retrospective study. The flowchart of study cohort selection was shown in Fig. 1. The characteristics of the study participants were displayed in Table 1. The median (quartiles) age of the patients was 63 (55, 69) years. The median (quartiles) follow-up time was 23 (3, 51) months. Forty-four patients developed ESRD. There was no significant difference in the medical history of CKD between ESRD and non-ESRD group. Patients who developed ESRD had poorer renal function with lower eGFR at the time of diagnosis and showed more frequent sclerotic class on kidney histopathology, compared with who did not developed ESRD. Patients with ESRD had lower platelet, alanine aminotransferase (ALT), aspartate aminotransferase (AST) and higher TC, TG, VLDL and procalcitonin (PCT) at diagnosis. Hypertension was more common in patients with ESRD. There were no significant
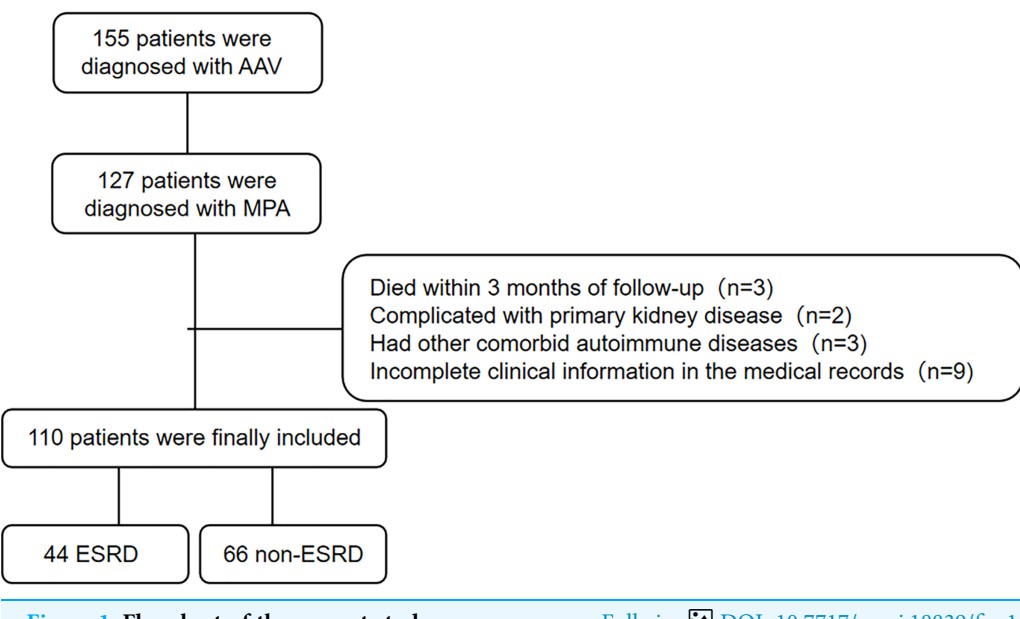

**Figure 1 Flowchart of the current study.**

**Table 1 Patient characteristics and comparisons made according to whether ESRD occurred.**

| | Total $n = 110$ | ESRD $n = 44$ | Non-ESRD $n = 66$ | P value |
|---|---|---|---|---|
| Age (year) | 63 (55, 69) | 62 (52, 68) | 64 (57, 70) | 0.180 |
| Gender (male) | 62 (56.4%) | 24 (54.5%) | 38 (57.6%) | 0.754 |
| BMI (kg/m$^2$) | 21.47 ± 3.10 | 21.22 ± 2.57 | 21.64 ± 3.423 | 0.094 |
| Hypertension | 60 (54.5%) | 32 (72.7%) | 28 (42.4%) | 0.002 |
| Diabetes | 18 (16.4%) | 6 (13.6%) | 12 (18.2%) | 0.528 |
| CKD | 3 (2.7%) | 2 (4.5%) | 1 (1.5%) | 0.563 |
| Hemoglobin (g/L) | 87.00 (71.00, 98.00) | 86.00 (71.25, 94.75) | 89.50 (71.00, 99.25) | 0.240 |
| WBC (×10$^9$/L) | 7.02 (5.32, 9.98) | 7.12 (4.96, 8.54) | 7.02 (5.50, 11.19) | 0.197 |
| N (%) | 78.25 (68.18, 85.63) | 78.45 (67.75, 83.73) | 77.90 (68.33, 86.50) | 0.903 |
| PLT (×10$^9$/L) | 254.50 (185.75, 335.75) | 209.50 (159.50, 290.00) | 276.00 (211.75, 374.25) | 0.004 |
| eGFR (mL/min) | 13.58 (7.10, 42.38) | 7.60 (5.78, 13.79) | 34.03 (14.60, 88.02) | <0.001 |
| Serum creatinine (μmmol/L) | 359.90 (129.75, 590.67) | 534.95 (425.67, 726.22) | 164.00 (81.95, 361.25) | <0.001 |
| BUN (mmol/L) | 18.98 (7.89, 29.33) | 28.43 (21.49, 39.66) | 10.71 (5.63, 20.88) | <0.001 |
| ALT (U/L) | 13.00 (8.00, 19.00) | 10.00 (7.00, 15.00) | 14.00 (8.00, 21.25) | 0.027 |
| AST (U/L) | 17.00 (14.00, 24.25) | 16.00 (12.25, 19.50) | 16.00 (12.25, 28.50) | 0.011 |
| Albumin (g/L) | 31.43 ± 5.05 | 31.56 ± 5.01 | 31.35 ± 5.12 | 0.831 |
| TC (mmol/L) | 4.08 (3.39, 4.88) | 4.50 (3.71, 5.42) | 3.93 (3.25, 4.60) | 0.015 |
| TG (mmol/L) | 1.32 (0.89, 1.84) | 1.61 (1.14, 2.14) | 1.13 (0.82, 1.59) | 0.001 |
| VLDL (mmol/L) | 0.61 (0.40, 0.84) | 0.74 (0.55, 0.95) | 0.51 (0.36, 0.74) | 0.001 |
| LDL (mmol/L) | 2.47 (1.94, 3.05) | 2.57 (1.83, 3.05) | 2.41 (1.94, 3.10) | 0.917 |
| HDL (mmol/L) | 0.84 (0.66, 1.21) | 0.86 (0.67, 1.28) | 0.83 (0.66, 1.20) | 0.963 |
| LDH (U/L) | 209.00 (171.00, 265.5) | 2,220.00 (163.50, 287.25) | 204.00 (173.50, 240.25) | 0.350 |

| | Total n = 110 | ESRD n = 44 | Non-ESRD n = 66 | P value |
|---|---|---|---|---|
| CRP (mg/L) | 26.75 (6.48, 59.20) | 14.33 (5.00, 46.54) | 31.37 (8.12, 61.15) | 0.110 |
| PCT (ng/mL) | 0.18 (0.07, 0.57) | 0.28 (0.11, 1.13) | 0.12 (0.05, 0.43) | 0.003 |
| D-dimer (mg/L) | 2.08 (0.92, 4.77) | 2.13 (1.12, 6.10) | 1.83 (0.85, 3.84) | 0.358 |
| PT (second) | 12.40 (11.25, 13.35) | 11.95 (11.10, 12.90) | 12.50 (11.35, 13.65) | 0.057 |
| APTT (second) | 30.70 (26.10, 36.80) | 29.30 (26.35, 34.88) | 30.90 (26.35, 37.25) | 0.257 |
| Kidney histopathology (Berden classification) | | | | |
| Focal | 8 (16.6%) | 0 (0%) | 8 (25%) | 0.002 |
| Crescentic | 17 (35.4%) | 4 (25%) | 13 (40.6%) | |
| Sclerotic | 14 (29.2%) | 10 (62.5%) | 4 (12.5%) | |
| Mixed | 9 (18.8%) | 2 (12.5%) | 7 (21.9%) | |
| Anti-MPO antibody (RU/mL) | 166.65 (106.53, 200.00) | 166.10 (102.33, 200.00) | 166.65 (108.18, 200.00) | 0.848 |
| BVAS | 18.00 (15.00, 26.00) | 19.00 (15.00, 27.75) | 16.00 (14.00, 24.25) | 0.082 |
| FFS | 2.00 (2.00, 3.00) | 2.00 (2.00, 3.00) | 2.00 (2.00, 3.00) | 0.520 |
| Immunosuppressive therapies | | | | |
| Cyclophosphamide | 80 (74.4%) | 35 (79.5%) | 45 (68.2%) | 0.190 |
| Rituximab | 14 (12.7%) | 6 (13.6%) | 8 (12.1%) | 0.815 |
| Lipid-lowering drugs | 18 (16.4%) | 8 (18.2%) | 10 (15.2%) | 0.674 |

Note:

Abbreviations: BMI, body mass index; CKD, chronic kidney disease; WBC, white blood cell; N, neutrophil; PLT, platelet; eGFR, estimated glomerular filtration rate; ALT, alanine aminotransferase; AST, aspartate aminotransferase; TC, total cholesterol; TG, triglycerides; VLDL, very low-density lipoprotein; LDL, low-density lipoprotein; HDL, high-density lipoprotein; LDH, lactate dehydrogenase; CRP, C-reactive protein; PCT, procalcitonin; PT, prothrombin time; APTT, activated partial thromboplastin time; MPO, myeloperoxidase; BVAS, Birmingham Vasculitis Activity Score; FFS, Five-Factor Score.

differences in age, gender, body mass index (BMI), other baseline examination results, anti-MPO antibody titers or immunosuppressive therapies (cyclophosphamide and rituximab).

As shown in Table 1, there were statistically significant differences in blood TG and VLDL at diagnosis between groups with or without ESRD in MPA patients. To further elucidate the differences between patients with high TG/VLDL and those with low TG/VLDL, we stratified the patient cohort into two groups based on the median value of TG/VLDL (Table 2). Patients with high TG/VLDL had worse renal function and lower eGFR at diagnosis. Hypertension is more common in patients with high TG/VLDL. Patients with high TG/VLDL had lower platelet (PLT) and ALT, and higher lactate dehydrogenase (LDH), PCT, and Five-Factor Score (FFS).

## Serum triglycerides at diagnosis were associated with ESRD development

Univariate Cox analysis showed that hypertension, higher TG, VLDL and lower PLT, eGFR and AST at diagnosis was associated with the development of ESRD (Table 3). After adjusted for age, gender, BMI, hypertension, diabetes, eGFR, and BVAS, TG at diagnosis remained to be associated with ESRD development in MPA patients (OR 1.230 95% CI

**Table 2 Patient characteristics and comparisons made according to high or low TG/VLDL.**

| | High TG n = 54 | Low TG n = 56 | P value | High VLDL n = 54 | Low VLDL n = 56 | P value |
|---|---|---|---|---|---|---|
| Age (year) | 62 (54, 71) | 64 (55, 68) | 0.971 | 64 (55, 71) | 63 (55, 68) | 0.558 |
| Gender (male) | 28 (51.9%) | 34 (60.7%) | 0.349 | 28 (51.9%) | 34 (60.7%) | 0.349 |
| BMI (kg/m$^2$) | 21.64 ± 2.79 | 21.32 ± 3.37 | 0.642 | 21.10 (19.83, 23.14) | 21.50 (18.81, 23.40) | 0.947 |
| Hypertension | 35 (64.8%) | 25 (44.6%) | 0.034 | 35 (64.8%) | 25 (44.6%) | 0.034 |
| Diabetes | 11 (20.4%) | 7 (12.5%) | 0.265 | 11 (20.4%) | 7 (12.5%) | 0.265 |
| CKD | 3 (5.6%) | 0 (0.0%) | 0.074 | 3 (5.6%) | 0 (0.0%) | 0.074 |
| Hemoglobin (g/L) | 84.20 ± 22.62 | 86.43 ± 19.69 | 0.583 | 82.33 ± 22.45 | 88.23 ± 19.50 | 0.144 |
| WBC (×10$^9$/L) | 7.07 (5.32, 9.62) | 6.84 (5.17, 10.90) | 0.929 | 7.27 (5.34, 9.76) | 6.64 (5.01, 10.65) | 0.441 |
| N (%) | 78.45 (70.88, 85.63) | 77.20 (66.85, 86.28) | 0.981 | 78.60 (71.92, 89.40) | 75.90 (65.95, 86.28) | 0.564 |
| PLT (×10$^9$/L) | 222.50 (160.50, 329.25) | 272.5 (206.75, 368.25) | 0.039 | 224.50 (160.50, 329.25) | 272.50 (206.75, 368.25) | 0.046 |
| eGFR (mL/min) | 9.95 (5.91, 21.04) | 23.64 (8.88, 90.22) | <0.001 | 9.95 (5.91, 19.92) | 27.44 (8.88, 90.22) | <0.001 |
| Serum creatinine (μmmol/L) | 431.05 (223.95, 701.53) | 243.40 (69.28, 478.68) | <0.001 | 431.65 (230.90, 706.73) | 212.00 (69.28, 478.68) | <0.001 |
| BUN (mmol/L) | 22.87 (14.34, 35.69) | 12.89 (5.40, 26.35) | <0.001 | 23.35 (14.81, 36.24) | 12.72 (5.40, 24.50) | <0.001 |
| ALT (U/L) | 10.50 (7.00, 15.00) | 15.00 (8.00, 24.50) | 0.007 | 10.50 (7.00, 15.00) | 14.50 (8.00, 24.50) | 0.009 |
| AST (U/L) | 16.00 (14.00, 22.25) | 18.00 (13.25, 25.75) | 0.263 | 16.00 (14.00, 22.25) | 17.50 (13.00, 25.75) | 0.485 |
| Albumin (g/L) | 31.30 ± 5.27 | 31.57 ± 4.89 | 0.780 | 30.88 ± 5.00 | 31.98 ± 5.10 | 0.260 |
| TC (mmol/L) | 4.24 (3.76, 5.41) | 3.68 (3.04, 4.48) | <0.001 | 4.21 (3.74, 5.41) | 3.73 (3.05, 4.48) | <0.001 |
| TG (mmol/L) | 1.84 (1.58, 2.39) | 0.90 (0.71, 1.10) | <0.001 | 1.84 (1.58, 2.39) | 0.90 (0.71, 1.12) | <0.001 |
| VLDL (mmol/L) | 0.84 (0.72, 1.07) | 0.40 (0.32, 0.51) | <0.001 | 0.85 (0.73, 1.07) | 0.40 (0.31, 0.51) | <0.001 |
| LDL (mmol/L) | 2.64 (2.01, 3.30) | 2.22 (1.84, 2.94) | 0.023 | 2.64 (1.98, 3.35) | 2.24 (1.86, 2.93) | 0.042 |
| HDL (mmol/L) | 0.78 (0.63, 1.01) | 0.92 (0.72, 1.37) | 0.081 | 0.77 (0.63, 0.95) | 0.97 (0.74, 1.40) | 0.014 |
| LDH (U/L) | 227.00 (188.50, 303.25) | 194.50 (167.25, 228.50) | 0.004 | 227.00 (191.75, 303.25) | 193.00 (167.00, 275.10) | 0.003 |
| CRP (mg/L) | 15.70 (5.00, 49.19) | 29.94 (8.11, 64.86) | 0.134 | 18.46 (5.00, 49.57) | 27.71 (7.28, 64.86) | 0.315 |
| PCT (ng/mL) | 0.29 (0.11, 0.78) | 0.14 (0.05, 0.31) | 0.026 | 0.40 (0.11, 0.79) | 0.10 (0.05, 0.27) | 0.003 |
| D-dimer (mg/L) | 2.18 (1.06, 5.50) | 1.75 (0.88, 3.39) | 0.192 | 2.22 (1.16, 7.07) | 1.53 (0.70, 2.92) | 0.014 |
| PT (second) | 11.95 (11.10, 13.03) | 12.50 (11.70, 13.70) | 0.052 | 11.95 (11.10, 13.03) | 12.50 (11.60, 13.70) | 0.062 |
| APTT (second) | 30.27 ± 8.50 | 32.45 ± 7.80 | 0.128 | 30.18 ± 8.45 | 32.54 ± 7.83 | 0.133 |
| Kidney histopathology (Berden classification) | | | | | | |
| Focal | 4 (16.7%) | 4 (16.7%) | 0.135 | 4 (15.3%) | 4 (18.2%) | 0.135 |
| Crescentic | 5 (20.8%) | 12 (50.0%) | | 6 (23.1%) | 11 (50%) | |
| Sclerotic | 10 (41.7%) | 4 (16.7%) | | 10 (38.5%) | 4 (18.2%) | |
| Mixed | 5 (20.8%) | 4 (16.7%) | | 6 (23.1%) | 3 (13.6%) | |
| Anti-MPO antibody | | | | | | |
| (RU/mL) | 188.40 (110.83, 200.00) | 152.55 (106.38, 200.00) | 0.34 | 181.50 (112.98, 200.00) | 160.30 (106.22, 200.00) | 0.425 |
| BVAS | 18.50 (15.00, 33.50) | 17.50 (14.00, 31.30) | 0.116 | 18.50 (15.00, 28.00) | 17.50 (14.00, 20.75) | 0.084 |
| FFS | 2.00 (2.00, 3.00) | 2.00 (1.00, 3.00) | 0.014 | 2.00 (2.00, 3.00) | 2.00 (1.00, 2.00) | 0.001 |
| Immunosuppressive therapies | | | | | | |
| Cyclophosphamide | 40 (74.1%) | 40 (71.4%) | 0.755 | 40 (74.1%) | 40 (71.4%) | 0.755 |

|  | High TG n = 54 | Low TG n = 56 | P value | High VLDL n = 54 | Low VLDL n = 56 | P value |
|---|---|---|---|---|---|---|
| Rituximab | 10 (18.5%) | 4 (7.1%) | 0.074 | 10 (18.5%) | 4 (7.1%) | 0.074 |
| Lipid-lowering drugs | 10 (18.5%) | 8 (14.3%) | 0.549 | 10 (18.5%) | 8 (14.3%) | 0.549 |

**Note:**
Abbreviations: BMI, body mass index; CKD, chronic kidney disease; WBC, white blood cell; N, neutrophil; PLT, platelet; eGFR, estimated glomerular filtration rate; ALT, alanine aminotransferase; AST, aspartate aminotransferase; TC, total cholesterol; TG, triglycerides; VLDL, very low-density lipoprotein; LDL, low-density lipoprotein; HDL, high-density lipoprotein; LDH, lactate dehydrogenase; CRP, C-reactive protein; PCT, procalcitonin; PT, prothrombin time; APTT, activated partial thromboplastin time; MPO, myeloperoxidase; BVAS, Birmingham Vasculitis Activity Score; FFS, Five-Factor Score.

**Table 3 Univariate COX proportional hazards model of ESRD.**

|  | HR (95%CI) | P value |
|---|---|---|
| Age (year) | 0.996 [0.975–1.017] | 0.701 |
| Gender (male) | 0.963 [0.531–1.746] | 0.901 |
| BMI (kg/m$^2$) | 0.972 [0.871–1.085] | 0.612 |
| Hypertension | 2.906 [1.487–5.681] | 0.002 |
| Diabetes | 0.802 [0.338–1.900] | 0.616 |
| Hemoglobin (g/L) | 0.978 [0.973–1.002] | 0.097 |
| WBC (×10$^9$/L) | 0.931 [0.857–1.012] | 0.092 |
| N (%) | 0.998 [0.972–1.024] | 0.879 |
| PLT (×10$^9$/L) | 0.996 [0.993–0.999] | 0.004 |
| ALT (U/L) | 0.979 [0.954–1.005] | 0.116 |
| AST (U/L) | 0.961 [0.927–0.996] | 0.030 |
| Albumin (g/L) | 1.007 [0.949–1.068] | 0.812 |
| eGFR (mL/min) | 0.900 [0.856–0.946] | <0.001 |
| TC (mmol/L) | 1.090 [0.901–1.319] | 0.375 |
| TG (mmol/L) | 1.268 [1.086–1.481] | 0.003 |
| VLDL (mmol/L) | 1.810 [1.295–2.529] | 0.001 |
| LDL (mmol/L) | 1.163 [0.873–1.549] | 0.303 |
| HDL (mmol/L) | 0.951 [0.499–1.811] | 0.878 |
| LDH (U/L) | 1.001 [0.999–1.004] | 0.238 |
| CRP (mg/L) | 0.995 [0.985–1.004] | 0.273 |
| PCT (ng/mL) | 1.018 [0.992–1.045] | 0.172 |
| D-dimer (mg/L) | 1.030 [0.975–1.088] | 0.294 |
| PT (second) | 0.939 [0.856–1.029] | 0.179 |
| APTT (second) | 0.981 [0.948–1.014] | 0.252 |
| Anti-MPO antibody (RU/mL) | 0.999 [0.994–1.003) | 0.560 |
| Kidney histopathology | 1.638 [0.981–2.733] | 0.059 |
| BVAS | 1.032 [1.000–1.065] | 0.051 |
| Cyclophosphamide | 0.682 [0.377–1.233] | 0.205 |
| Rituximab | 0.375 [0.051–2.734] | 0.333 |
| Lipid-lowering drugs | 1.117 [0.518–2.409] | 0.778 |

**Note:**
Abbreviations: BMI, body mass index; WBC, white blood cell; N, neutrophil; PLT, platelet; ALT, alanine aminotransferase; AST, aspartate aminotransferase; TC, total cholesterol; TG, triglycerides; VLDL, very low-density lipoprotein; LDL, low-density lipoprotein; HDL, high-density lipoprotein; LDH, lactate dehydrogenase; CRP, C-reactive protein; PCT, procalcitonin; PT, prothrombin time; APTT, activated partial thromboplastin time; MPO, myeloperoxidase; BVAS, Birmingham Vasculitis Activity Score; HR, hazard ratio; CI, confident interval.

**Table 4 Multivariable COX proportional hazards model of ESRD.**

| | OR (95%CI) | P value | | OR (95%CI) | P value |
|---|---|---|---|---|---|
| Age (year) | 0.971 [0.935–1.008] | 0.122 | Age (year) | 0.969 [0.934–1.005] | 0.095 |
| Gender (male) | 1.221 [0.574–2.599] | 0.604 | Gender (male) | 1.171 [0.557–2.465] | 0.677 |
| BMI (kg/m$^2$) | 0.991 [0.861–1.140] | 0.899 | BMI (kg/m$^2$) | 0.991 [0.861–1.141] | 0.903 |
| Hypertension | 1.073 [0.406–2.838] | 0.886 | Hypertension | 1.063 [0.403–2.807] | 0.901 |
| Diabetes | 0.620 [0.172–2.237] | 0.466 | Diabetes | 0.647 [0.181–2.316] | 0.504 |
| eGFR (mL/min) | 0.881 [0.822–0.945] | <0.001 | eGFR (mL/min) | 0.882 [0.823–0.946] | <0.001 |
| BVAS | 0.967 [0.914–1.023] | 0.237 | BVAS | 0.967 [0.914–1.024] | 0.25 |
| TG (mmol/L) | 1.230 [1.009–1.498] | 0.040 | VLDL (mmol/L) | 1.506 [0.957–2.368] | 0.076 |
| Age (year) | 0.972 [0.937–1.008] | 0.127 | Age (year) | 0.967 [0.933–1.003] | 0.071 |
| Gender (male) | 1.206 [0.579–2.511] | 0.617 | Gender (male) | 1.021 [0.502–2.077] | 0.954 |
| BMI (kg/m$^2$) | 0.975 [0.844–1.125] | 0.725 | BMI (kg/m$^2$) | 0.978 [0.849–1.125] | 0.752 |
| Hypertension | 1.184 [0.444–3.155] | 0.735 | Hypertension | 1.153 [0.439–3.031] | 0.773 |
| Diabetes | 0.584 [0.158–2.159] | 0.420 | Diabetes | 0.632 [0.177–2.256] | 0.480 |
| eGFR (mL/min) | 0.885 [0.828–0.947] | <0.001 | eGFR (mL/min) | 0.886 [0.829–0.946] | <0.001 |
| BVAS | 0.971 [0.923–1.022] | 0.266 | BVAS | 0.975 [0.926–1.027] | 0.343 |
| TC (mmol/L) | 1.274 [0.964–1.682] | 0.089 | LDL (mmol/L) | 1.302 [0.934–1.815] | 0.119 |

**Note:**
**Abbreviations:** BMI, body mass index; eGFR, estimated glomerular filtration rate; BVAS, Birmingham Vasculitis Activity Score; TG, triglycerides; VLDL, very low-density lipoprotein; TC, total cholesterol; LDL, low-density lipoprotein; OR, odds ratio; CI, confidence interval.

[1.009–1.498], $P = 0.040$). VLDL demonstrated a marginal trend towards association with ESRD development (OR 1.506 95% CI [0.957–2.368], $P = 0.076$) (Table 4).

In order to further evaluate the relationship between blood lipid levels and renal prognosis in MPA patients, we employed the ROC curve to evaluate the sensitivity and specificity of the association between blood lipid levels and the occurrence of ESRD in MPA patients. Based on the Youden's index (sensitivity+specificity-1), the optimal cutoff values for blood TG, VLDL, TC, and LDL were determined to be 1.45, 0.66, 4.59, and 2.54 mmol/L, respectively. According to this cutoff value, the patients were divided into two groups. Survival analysis revealed that patients with TG > 1.45 mmol/L or VLDL > 0.66 mmol/L had significantly higher risk of ESRD development than those with TG ≤ 1.45 mmol/L or VLDL ≤ 0.66 mmol/L ($P = 0.0004$, $P = 0.0002$) (Fig. 2). Patients with TC > 4.59 mmol/L also showed higher ESRD risk ($P = 0.0058$). Patients with VLDL > 0.73 mmol/L had significantly all-cause mortality ($P = 0.0368$) (Fig. 3).

## Serum triglycerides and VLDL at diagnosis were associated with severe renal involvement and disease activity

Correlation analysis was performed between serum TG or VLDL levels and serum creatinine, urine volume, urine albumin creatine ratio (UACR), urine erythrocyte count, BVAS and anti-MPO antibody titers at diagnosis. Serum TG or VLDL positively correlated with serum creatinine and urine erythrocyte count, and inversely correlated with urine volume (Table 5). There was also a trend of positive correlation between serum TG or VLDL and anti-MPO antibody titers and BVAS. The results indicated that higher TG or

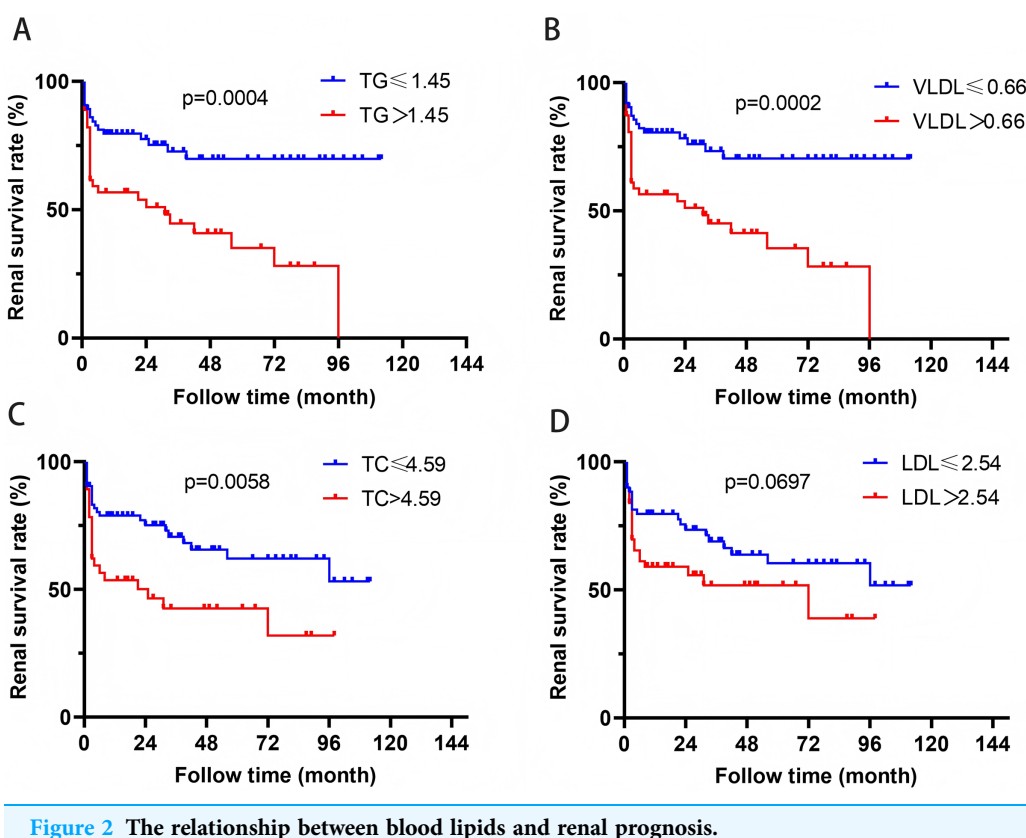

**Figure 2** The relationship between blood lipids and renal prognosis.

VLDL was associated with advanced renal impairment and high disease activity at the time of diagnosis.

## High triglycerides possibly mediated proinflammatory and profibrotic process

To further explore the mechanism underlying the poorer renal outcome in patients with high TG level, DIA quantification proteomics analysis of plasma samples was performed. Ten patients newly diagnosed with MPA in our hospital were prospectively enrolled in the study and fasting blood samples before treatments were collected. According to the serum TG at diagnosis, these patients were divided into two groups: high-TG ($n = 5$) and low-TG ($n = 5$). None of these patients had a history of CKD. The high-TG group showed higher BMI, anti-MPO antibody titers and lower eGFR than the low-TG group (Table 6). Principal component analysis (PCA) separated the samples of patients with high TG level from those with low TG level based upon their proteomic profiles (Fig. 4A). High TG group and low TG group exhibited markedly distinct proteomic profiles, with the most prominently differentially expressed proteins depicted in Fig. 4B. Several profibrotic pathways were identified using GO or KEGG analysis of upregulated proteins in high TG group, which included extracellular matrix organization and growth factor binding (Fig. 4C). Compared with the low TG group, the high TG group showed profound inflammatory responses, as evidenced by the upregulation of NF-κB, Toll-like and TNF

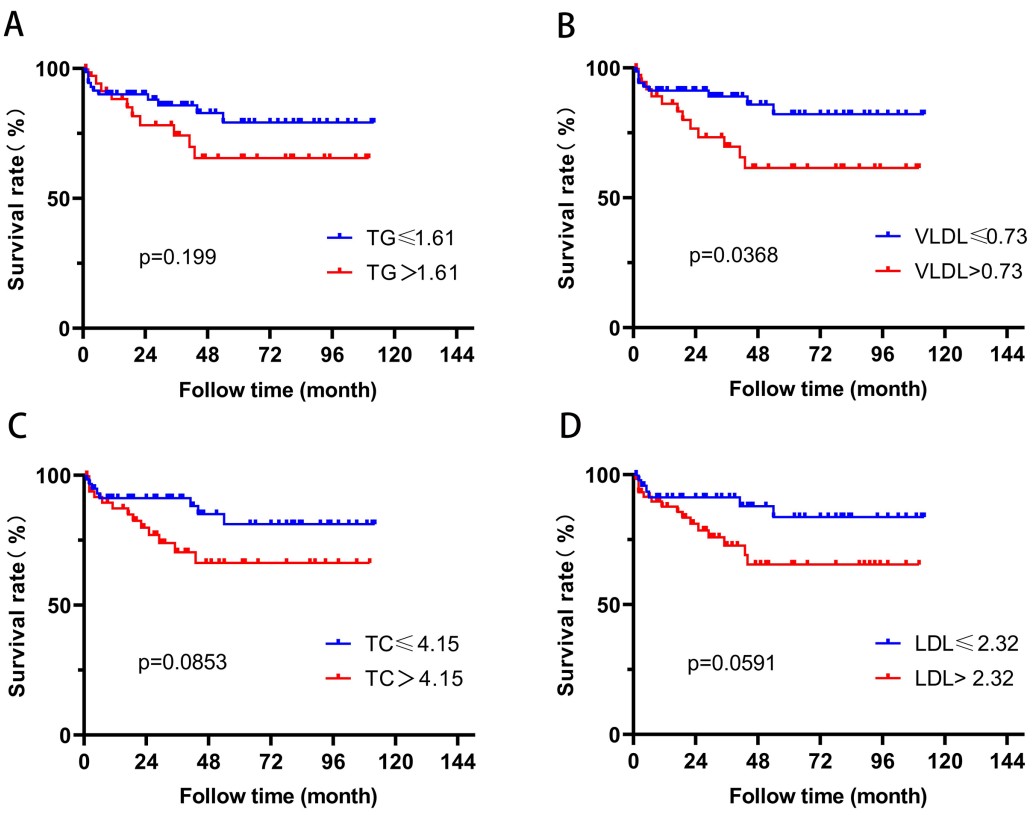

**Figure 3 The relationship between blood lipids and mortality.**

**Table 5 The association between admission TG/VLDL and renal involvement and disease activity.**

|  | Spearman R | *P* value |
| --- | --- | --- |
| **TG** |  |  |
| Serum creatinine (µmmol/L) | 0.422 | <0.001 |
| Urine volume (mL) | −0.299 | 0.004 |
| UACR (mg/g) | 0.157 | 0.246 |
| Urine erythrocyte count (/uL) | 0.207 | 0.034 |
| BVAS | 0.161 | 0.093 |
| Anti-MPO antibody (RU/mL) | 0.177 | 0.064 |
| **VLDL** |  |  |
| Serum creatinine (µmmol/L) | 0.448 | <0.001 |
| Urine volume (mL) | −0.307 | 0.003 |
| UACR (mg/g) | 0.153 | 0.260 |
| Urine erythrocyte count (/uL) | 0.207 | 0.034 |
| BVAS | 0.176 | 0.065 |
| Anti-MPO antibody (RU/mL) | 0.165 | 0.085 |

Note:
**Abbreviations:** TG, triglycerides; UACR, urine albumin creatine ratio; MPO, myeloperoxidase; BVAS, Birmingham Vasculitis Activity Score; VLDL, very low-density lipoprotein.

**Table 6 Characteristics of 10 patients included in DIA quantitative proteomics analysis.**

|  | High TG<br>n = 5 | Low TG<br>n = 5 |
|---|---|---|
| **Baseline** |  |  |
| Age (year) | 56 ± 12 | 61 ± 11 |
| Gender (male) | 3 (60%) | 2 (40%) |
| BMI (kg/m²) | 22.98 ± 2.72 | 21.15 ± 4.11 |
| Hypertension | 3 (60%) | 3 (60%) |
| Diabetes | 0 (0%) | 1 (20%) |
| CKD | 0 (0%) | 0 (0%) |
| eGFR (mL/min) | 15.00 ± 10.90 | 35.08 ± 15.21 |
| TC (mmol/L) | 5.53 ± 4.04 | 4.53 ± 0.38 |
| TG (mmol/L) | 2.50 ± 1.47 | 1.00 ± 0.45 |
| VLDL (mmol/L) | 1.16 ± 0.66 | 0.46 ± 0.21 |
| LDL (mmol/L) | 3.60 ± 2.75 | 2.92 ± 0.32 |
| HDL (mmol/L) | 0.91 ± 0.46 | 1.24 ± 0.43 |
| Anti-MPO antibody (RU/mL) | 177.16 ± 22.97 | 109.06 ± 89.55 |
| BVAS | 18.00 ± 2.12 | 8.40 ± 1.52 |
| Kidney histopathology (Berden classification) |  |  |
| Focal | 0 (0%) | 2 (40%) |
| Crescentic | 2 (40%) | 2 (40%) |
| Sclerotic | 1 (20%) | 0 (0%) |
| Mixed | 0 (0%) | 0 (0%) |
| Immunosuppressive therapies |  |  |
| Cyclophosphamide | 2 (40%) | 2 (40%) |
| Rituximab | 3 (60%) | 2 (40%) |
| **Follow-up** |  |  |
| ESRD | 3 | 1 |
| Dependent dialysis | 3 | 0 |

**Note:**
 **Abbreviations:** CKD, chronic kidney disease; eGFR, estimated glomerular filtration rate; TC, total cholesterol; TG, triglycerides; VLDL, very low-density lipoprotein; LDL, low-density lipoprotein; HDL, high-density lipoprotein; PCT, procalcitonin; MPO, myeloperoxidase; BVAS, Birmingham Vasculitis Activity Score; ESRD, end-stage renal disease.

signaling pathway, which indicated that high TG possibly promoted inflammation (Fig. 4D). The high TG group also exhibited upregulated complement and coagulation cascades (Fig. 4D). We conducted a follow-up study on these 10 patients through July 2024, observing that 60% of the patients in the high TG group had progressed to ESRD and initiated dialysis treatment. In contrast, only 20% of the patients in the low TG group reached ESRD (Table 6).

## DISCUSSION

MPA is a member of AAV characterized by vascular endothelial inflammatory destruction mediated by p-ANCA, with a kidney involvement rate of up to 100%, leading to a high risk of progression to ESRD (*Jennette, 2013*; *Moiseev et al., 2016*; *Pope et al., 2023*; *Redondo-Rodriguez et al., 2022*). Dyslipidemia is a critical contributor in endothelial dysfunction. In

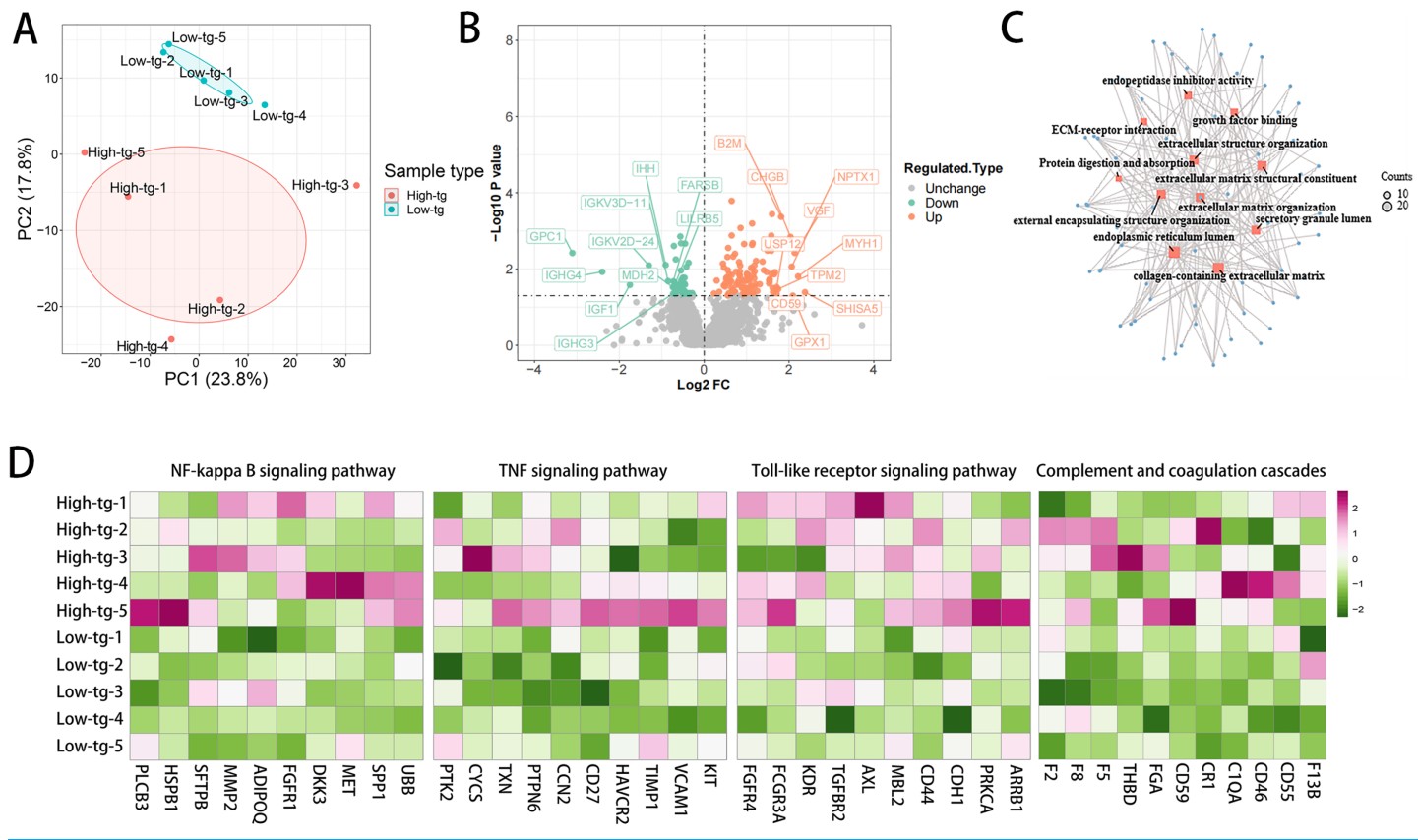

**Figure 4 The high TG group showed upregulated profibrotic, proinflammatory signaling pathways and complement and coagulation cascades.** (A) Principal component analysis (PCA) separates the proteomic profiles of patients with high and low TG; (B) Volcano plot of differentially expressed proteins; (C) The profibrotic pathway was upregulated in the high TG group. (D) Inflammatory signaling pathways and complement and coagulation cascades were upregulated in the high TG group.

the present study, we explored the association of dyslipidemia at the time of diagnosis with the renal outcome of MPA patients and demonstrated that in MPA, high TG or VLDL at diagnosis was associated with increased risk of ESRD development. DIA quantification proteomics analysis showed that patients with high TG levels and severe MPA had up-regulated profibrotic pathway, inflammatory signaling pathways and complement and coagulation cascades compared with those with lower TG levels and milder disease severity, which were possibly associated with poor renal prognosis. *Szeto et al. (2016)* found that high-fat diet could induce mitochondrial damage in glomerular endothelial cells, podocytes, and proximal tubular epithelial cells in C57BL/6 mice, showing loss of glomerular endothelial cells and podocytes, mesangial expansion, glomerulosclerosis, macrophage infiltration, and upregulation of proinflammatory and profibrotic cytokines. *Yamamoto et al. (2017)* showed that high-fat diet resulted in kidney injury by causing lysosomal dysfunction and autophagic flux impairment. *Wallace et al. (2019)* demonstrated that lipid levels increased during remission induction among patients with newly-diagnosed AAV and PR3-ANCA positive but not among those with MPO-ANCA positive. The sequential monitoring of blood lipids during the therapeutic process were not

analyzed in our study and the significance of lipid level changes in the prognosis of MPA remains to be investigated in further study.

Renal fibrosis is characterized by excessive deposition of extracellular matrix leading to scarring, which is a characteristic manifestation of ESRD (*Huang, Fu & Ma, 2023*). Our study showed upregulated pathways associated with extracellular matrix formation in patients in high TG group. *Jiang et al. (2005)* found that hyperlipidemia could increase renal lipid accumulation in C57BL/6j mice through the sterol regulatory element-binding protein-1c dependent pathway and upregulate the expression of plasminogen activator inhibitor-1 (PAI-1), vascular endothelial growth factor (VEGF), and extracellular matrix (such as type IV collagen and fibronectin) in the kidneys, leading to glomerulosclerosis. *Liu et al. (2023)* illustrated that a high-fat diet induced hyperglycemia, hyperinsulinemia and hypertriglyceridemia, leading to lipid droplet deposit in renal tubular cells and interstitial extracellular matrix accumulation through sterol regulatory element binding protein-1 (SREBP-1) and transforming growth factor-β1 (TGF-β1) (*Hao et al., 2012*). TGF-β1 was a major driver of renal fibrosis that could stimulate the production of extracellular matrix by primarily activating the Smad pathway in renal tubular cells and fibroblasts, thereby promoting renal tubular cell damage, transformation, and fibrosis (*Hao et al., 2012*; *Meng, Nikolic-Paterson & Lan, 2016*; *Yuan, Tang & Zhang, 2022*). *Yang et al. (2022)* demonstrated that very-low-density lipoprotein receptor (VLDLR)-enhanced lipid metabolism in pancreatic stellate cells (PSCs) was a critical driver of fibrotic progression in chronic pancreatitis. These results were consistent with our findings, indicating that high TG could possibly promote extracellular matrix formation and lead to renal fibrosis.

Our study suggested that high TG was linked with upregulated inflammatory response, including NF-κB, Toll-like and TNF signaling pathway. The Toll-like receptors (TLR) family, evolutionarily conserved across species, functions by detecting highly conserved structural motifs, which include both pathogen-associated molecular patterns (PAMPs) and danger-associated molecular patterns (DAMPs) (*Arleevskaya et al., 2020*). Stimulation of TLR by PAMPs or DAMPs can activate NF-κB and mitogen activated protein kinase (MAPK) pathways, then inducing the production of pro-inflammatory cytokines (such as TNF-α, IL-1, IL-12) (*Arleevskaya et al., 2020*). An *in vitro* study showed that oxLDL could upregulate TLR-4 in macrophages (*Xu et al., 2001*). An animal experiment found that fenofibrate, an approved agent for dyslipidemia, could inhibit NF-κB expression in diabetes nephropathy (*Chen et al., 2008*). *Tan et al.'s (2022)* study identified a pro-inflammatory role of TG in Behçet's disease and experimental autoimmune uveitis, where TGs stimulated CD4+ T cell activation and Th17 differentiation, increasing inflammation. The inhibition of TG generation reduced disease severity, suggesting TG as a therapeutic target for inflammation in uveitis.

In our study, we found that the high TG group exhibited activated complement and coagulation cascades compared with those in low TG group. The complement system is crucial for innate immunity, involved in immune complex clearance, angiogenesis, and lipid metabolism (*Ricklin et al., 2010*). Complement system activation is involved in various kidney diseases including ANCA-associated kidney injury (*Deng, Gao & Zhao, 2022*; *Xu, Tao & Su, 2022*). The activation of the alternative complement pathway leads to

the production of C5a, which is a component of the pathogenesis of ANCA related vasculitis (*Tesar & Hruskova, 2018*; *Xiao et al., 2014*). The latest therapeutic drug, avacopon, a C5a receptor inhibitor, has been approved for the treatment of AAV (*Lee, 2022*; *Tesar & Hruskova, 2018*). In addition, the inhibition of complement C5a using NOX-D21 has been shown to alleviate renal fibrosis and enhance lipid metabolism in diabetic nephropathy (*Yiu et al., 2018*). The complement data were not collected in our study. Other studies have found that low C3 was associated with poor renal prognosis in AAV (*Augusto et al., 2016*; *Crnogorac et al., 2018*). Lipid disorder has been reported to relate with the complement system. A high-fat diet can promote the production of pro-inflammatory cytokines, chemokines, and cell adhesion molecules, by upregulating the expression of C3aR and C5aR on macrophages and activating M1, then stimulating kidney damage and fibrosis (*Mamane et al., 2009*; *Phieler et al., 2013*; *Xu, Tao & Su, 2022*). Targeting C3aR and C5aR can inhibit obesity induced by diet and the signal transduction of macrophages (*Lim et al., 2013*). The complement also contributes to the homeostasis of lipid metabolism (*Corvillo & Akinci, 2019*). Another study showed that the alternative complement pathway significantly contributed to atherosclerosis in LDL receptor-deficient mice, particularly under conditions of endotoxin exposure or high-fat diet, and is associated with altered lipid metabolism (*Malik et al., 2010*). In addition, CKD patients usually experience coagulation abnormalities, which possibly result from endothelial damage and inflammation (*Lutz et al., 2014*). Tissue factor III (TF) activates the extrinsic coagulation system upon vascular injury and is implicated in multiple kidney diseases (*Madhusudhan, Kerlin & Isermann, 2016*; *Oe & Takahashi, 2022*). Research reported that under the condition of dyslipidemia, TF and its downstream coagulation proteases could activate protease-activated receptors (PARs), then activate NF-κB or MAPK, ultimately exacerbating inflammation and fibrosis, leading to the progression of kidney damage (*Oe & Takahashi, 2022*). In diabetes nephropathy, thrombin could lead to glomerular fibrin and extracellular matrix aggregation by inducing mesangial TGF-β expression and elevating the PAI-1 to tPA ratio in peripheral blood (*Madhusudhan, Kerlin & Isermann, 2016*). *Kim et al. (2015)* reported that elevated blood lipids were associated with increased coagulation activity in a normal population, as evidenced by shorter prothrombin time and altered thrombin generation assay values.

## Limitations

There were several limitations in the present study. Firstly, this is a single center study with a small number of cases, and it needs to be validated in other centers. Secondly, the blood lipid levels in our study were measured at diagnosis and may have been influenced by MPA or MPA-induced eGFR reduction. We did not analyze the relationship between pre-onset lipid levels and MPA prognosis, a gap that warrants exploration in future research to inform potential lipid management strategies prior to disease onset. Thirdly, in the proteomics part, while TG levels were associated with the activation of certain pathways, these results may also be affected by the activity of MPA. The existing proteomic data serve only as a preliminary exploration of the potential biological mechanisms in patients with high TG levels. Future studies may require the integration of more comprehensive clinical

data with proteomic analysis for joint modeling to explore the independent contributions of different factors.

## CONCLUSION

In MPA patients, high TG or VLDL at diagnosis is associated with an increased risk of ESRD development. The underlying mechanisms are possibly be associated with the upregulation of pro-fibrotic and inflammatory signaling pathways, and the activation of complement and coagulation cascade reactions. Our research may lead clinical doctors to pay more attention to the blood lipid status of MPA patients during the diagnosis and treatment process.

### Funding

This work was supported by grants from National Natural Science Foundation of China (No. 82070720 and No. 82300806), Young and Middle-aged Scientific Research Major Project of Fujian Provincial Health Commission (No. 2021ZQNZD004), Joint Funds for the Innovation of Science and Technology of Fujian province (2021Y9100). The funders had no role in study design, data collection and analysis, decision to publish, or preparation of the manuscript.

### Grant Disclosures

The following grant information was disclosed by the authors:
National Natural Science Foundation of China: 82070720 and 82300806.
Young and Middle-Aged Scientific Research Major Project of Fujian Provincial Health Commission: 2021ZQNZD004.
Joint Funds for the Innovation of Science and Technology of Fujian Province: 2021Y9100.

### Competing Interests

The authors declare that they have no competing interests.

### Author Contributions

- Zigui Zheng conceived and designed the experiments, performed the experiments, analyzed the data, prepared figures and/or tables, authored or reviewed drafts of the article, and approved the final draft.
- Yujia Wang conceived and designed the experiments, performed the experiments, analyzed the data, prepared figures and/or tables, authored or reviewed drafts of the article, and approved the final draft.
- Jingzhi Xie performed the experiments, prepared figures and/or tables, and approved the final draft.
- Zhimin Chen analyzed the data, prepared figures and/or tables, and approved the final draft.
- Bingjing Jiang performed the experiments, prepared figures and/or tables, and approved the final draft.

- Yanfang Xu conceived and designed the experiments, authored or reviewed drafts of the article, and approved the final draft.

## Human Ethics

The following information was supplied relating to ethical approvals (*i.e.*, approving body and any reference numbers):

The Ethics Committee of the First Affiliated Hospital of Fujian Medical University (protocal code [2023]-180).

## Data Availability

The DIA quantification proteomics data are available at ProteomeXchange: PXD045969.

The raw data is available in the Supplemental Files.

## Supplemental Information

Supplemental information for this article can be found online at http://dx.doi.org/10.7717/peerj.18839#supplemental-information.

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
