# Peer review of "The association between serum lipids at diagnosis and renal outcome in microscopic polyangiitis patients"

_PeerJ, doi:10.7717/peerj.18839_

## Round 0.1 · original submission · Major Revisions

Some issues regarding the model of multivariate analysis exist and the authors should resolve it. Important factors which contribute to the outcome of a patient with vasculitis were not included in the model, and this is a concern for the validity of the reported findings.

·

Basic reporting

Clear, unambiguous, professional English language used throughout. -Yes
Intro & background to show context. Literature well referenced & relevant. -Yes
Structure conforms to PeerJ standards, discipline norm, or improved for clarity. -Yes
Figures are relevant, high quality, well labelled & described. -Yes
Raw data supplied (see PeerJ policy). -Yes

The authors conducted a retrospective cohort study to clarify the association of dyslipidemia with renal prognosis among patients who developed microscopic polyangiitis (MPA). They revealed that high triglycerides (TG) or high very low-density lipoprotein (VLDL) level (described as TG/VLDL level below) at diagnosis is significantly associated with development of end stage renal disease (ESRD). They found the same results in the additional small (n=10) but prospective cohort. Lastly, they conducted proteomics analysis using serum samples of the patients and indicated that profibrotic, inflammatory, complement, and coagulation cascades are more accelerated in patients with high TG/VLDL level at diagnosis. Based on those results, they concluded that high TG/VLDL level is associated with an increased risk of ESRD among patients with MPA through the activation of profibrotic, inflammatory, complement, and coagulation cascades. The manuscript is basically well written and their research theme and findings have much curiosity. However, there are several questions about their experimental design especially at the retrospective analysis part.

Experimental design

Original primary research within Scope of the journal. -Yes
Research question well defined, relevant & meaningful. It is stated how the research fills an identified knowledge gap. -Yes
Rigorous investigation performed to a high technical & ethical standard. -Yes
Methods described with sufficient detail & information to replicate. -Yes

Experimental design is well constructed to clarify their hypothesis about the association of dyslipidemia with renal outcome on MPA. However, I have several questions about their methodology at the retrospective analysis part. I assume they need to modify/add the analysis and description to prove that high TG/VLDL level really contributed to ESRD development.

At table 1, authors compared patients who developed ESRD with those did not develop ESRD. However, to evaluate whether high TG/VLDL level contributes to ESRD development, high and low TG/VLDL patients should be compared. Authors should answer to this question.

Authors analyzed the association of TG/VLDL levels at diagnosis, not at/before disease onset, with the renal outcome. TG/VLDL at diagnosis might have been largely affected by MPA or MPA-induced eGFR decrease. Therefore, authors should mention the gap of the values between at/before disease onset and at diagnosis. Additionally, authors mentioned TG/HDL levels at diagnosis as “baseline” values in line 3, 41, 187, 213, and 318, and those should be described with other words to avoid reader’s misunderstanding.

To address whether TG/VLDL level at diagnosis really contribute to ESRD development, the effect of confounders which affects both TG/VLDL level at diagnosis and ESRD development should be adjusted with multivariate analysis. For example, severity of vasculitis may largely affect both TG/VLDL level at diagnosis and ESRD development. Additionally, age, gender, BMI, diabetes, hypertension, medication, eGFR at diagnosis, disease activity on renal biopsy and so on would be the possible confounders. Although authors adjusted for such as PLT and PCT (and did not adjust for gender), those seem not be the representative confounders which affect both TG/VLDL level at diagnosis and ESRD development. Multivariate analysis adjusting for representative confounders should be taken into account and authors are encouraged to describe the reason for their choice about the adjusted confounders.

Same questions, the gap between at/before disease onset and at diagnosis and adjustment for confounders, are also raised in the proteomics part. Activation of the pathways among high TG/VLDL patients could be caused by severity of MPA not by high TG/VLDL level at diagnosis. Authors should mention about the issue also in the proteomics section.

Validity of the findings

Impact and novelty is not assessed. Meaningful replication encouraged where rationale & benefit to literature is clearly stated. -Yes
All underlying data have been provided; they are robust, statistically sound, & controlled. -Yes
Conclusions are well stated, linked to original research question & limited to supporting results. -No

Their study and results have enough validity except for the parts I mentioned in EXPERIMENTAL DESIGN section.

Additional comments

I have several minor questions.

BMI should be taken in the analyses, if possible.

How about the result of Cox regression analysis comparing high and low TG/VLDL patients?

Why patients were divided to two groups with the values like TG on 1.66 mmol/L or HDL on 8.88 mmol/L? Are those mean or median values?

How about the result of eGFR or renal biopsy finding in table 2?

It might be better to simplify the description about introduction of previous studies in the discussion part.

Reviewer 2 ·

Basic reporting

The authors investigated the association between lipid levels and kidney prognosis in AAV patients. In addition, they applied proteomics methods to try to determine the mechanism of this association. The study itself is interesting and focused on a good topic to investigate; however, there are several important concerns.

I have some questions about "methods" regarding proteomics and analyses for the outcomes.

Experimental design

Method and study design
Why did the authors not include GPA patients as AAV in this study?

The authors need to show the patients’ selection as a figure including how many patients were excluded based on the exclusion criteria.

In addition, how many patients died within three months? And were there any patients with ESKD? Because “death” is the competitive outcome of ESKD.

Validity of the findings

Regarding the choice of the factors in the multivariate analysis, it seemed that the authors selected variables based on the P-values in the univariate analysis. However, this is not a good way to build the multivariate model. Please see the web https://www.acpjournals.org/journal/aim/authors/statistical-guidance#multivariable-analyses
(this is the statistical guidance issued by Annals of Internal Medicine)

Regarding proteomics analyses, how many plates were used in the experiment? I mean.. did the authors perform batch correction for the values?

Results
Please replace P=0.000 with P<0.001 in the tables.

It is essential to demonstrate and consider the effect of lipid-lowering drugs.
This cannot be allowed by just referring as a limitation.

It would be helpful to add volcano plots based on the results from proteomics analysis. In addition, please label or indicate the specific proteins detected by proteomics analyses.

Additional comments

Regarding the Discussion:
Why are “upregulation of pro-fibrotic and inflammatory signaling pathways, and the activation of complement and coagulation cascade” associated with the high level of TG?

---

## Round 0.2 · accepted · Accept

All the reviewers' concerns were resolved in this revised version.

·

Basic reporting

They revised the manuscript very well.
No additional requirement for revision.

Experimental design

No additional requirement for revision.

Validity of the findings

No additional requirement for revision.

Reviewer 2 ·

Basic reporting

The authors addressed my concerns and the quality of this manuscript was improved.

Experimental design

Statistical part is revised nicely, I think.

Validity of the findings

None.

Additional comments

The authors addressed my concerns and the quality of this manuscript was improved.